# Ketones from aldehydes via alkyl C(sp$^3$)−H functionalization under photoredox cooperative NHC/palladium catalysis

Hai-Ying Wang[1,2], Xin-Han Wang[1,2], Bang-An Zhou[1,2], Chun-Lin Zhang [1]✉ & Song Ye [1,2]✉

Direct synthesis of ketones from aldehydes features high atom- and step-economy. Yet, the coupling of aldehydes with unactivated alkyl C(sp$^3$)-H remains challenging. Herein, we develop the synthesis of ketones from aldehydes via alkyl C(sp$^3$)-H functionalization under photoredox cooperative NHC/Pd catalysis. The two-component reaction of iodomethylsilyl alkyl ether with aldehydes gave a variety of β-, γ- and δ-silyloxylketones via 1,n-HAT ($n$ = 5, 6, 7) of silylmethyl radicals to generate secondary or tertiary alkyl radicals and following coupling with ketyl radicals from aldehydes under photoredox NHC catalysis. The three-component reaction with the addition of styrenes gave the corresponding ε-hydroxylketones via the generation of benzylic radicals by the addition of alkyl radicals to styrenes and following coupling with ketyl radicals. This work demonstrates the generation of ketyl radical and alkyl radical under the photoredox cooperative NHC/Pd catalysis, and provides two and three component reactions for the synthesis of ketones from aldehydes with alkyl C(sp$^3$)-H functionalization. The synthetic potential of this protocol was also further illustrated by the late-stage functionalization of natural products.

Being one of the most basic functional groups with remarkable reactivity, ketones play a fundamental role in synthetic organic chemistry. In addition, they are also important and ubiquitous motifs in natural products, pharmaceuticals and functional materials[1–4]. Classic approaches to ketones include the oxidation of alcohols and acylation reaction with activated acyl electrophiles (Fig. 1a), such as acyl chlorides, amides, anhydrides and thioesters, via Friedel-Crafts acylation[5], substitution with organometallic reagents[6,7], and transition metal-catalyzed coupling with organostannanes, boronic esters, and halides[8–10]. In modern organic synthesis, direct acylation with aldehydes is an alternative and attractive approach to ketones, featuring high atom- and step-economy[11]. Most acylation reactions with aldehydes rely on transition metal (TM) catalysis (Fig. 1b). The TM-catalyzed coupling of aldehydes with organoboronic reagents[12–14],

hypervalent iodine reagents[15,16] and aryl/alkyl halides[17–19] has been well established. In addition, the TM-catalyzed aryl-H acylation with aldehydes were also reported[20–23]. Recently, the pioneering acylation of benzylic C(sp$^3$)-H bonds was developed via photoredox Ni catalysis[24]. The direct acylation of unactivated alkyl C(sp$^3$)-H bonds with aldehydes is one of the most appealing approaches for ketone synthesis but far less developed.

In the past few decades, N-heterocyclic carbenes (NHCs) have emerged as powerful organocatalysts to construct structurally diverse molecules[25–31]. NHC-catalyzed reactions of aldehydes with carbonyl compounds (benzoin reaction), imines (aza-benzoin reaction), and activated alkenes (Stetter reaction) have been well established for the synthesis of functionalized ketones via the umpolung of aldehydes[32,33]. In addition, NHC catalysis via radicals has opened up new avenues for

[1]Beijing National Laboratory for Molecular Sciences, CAS Key Laboratory of Molecular Recognition and Function, CAS Research/Education Center for Excellence in Molecular Sciences, Institute of Chemistry, Chinese Academy of Sciences, 100190 Beijing, China. [2]University of Chinese Academy of Sciences, 100049 Beijing, China. ✉e-mail: zhangchunlin@iccas.ac.cn; songye@iccas.ac.cn

**a Classical Ketone Synthesis from Alcohols and Acyl Electrophiles**

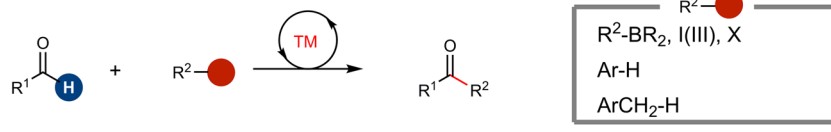

❖ **Modern Ketone Synthesis from Aldehydes (b-d)**

**b via TM-catalyzed Coupling Reaction**

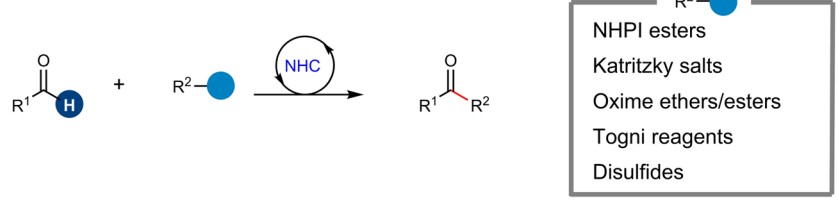

**c via NHC-catalyzed Acylation with Activated Alkylation Reagents**

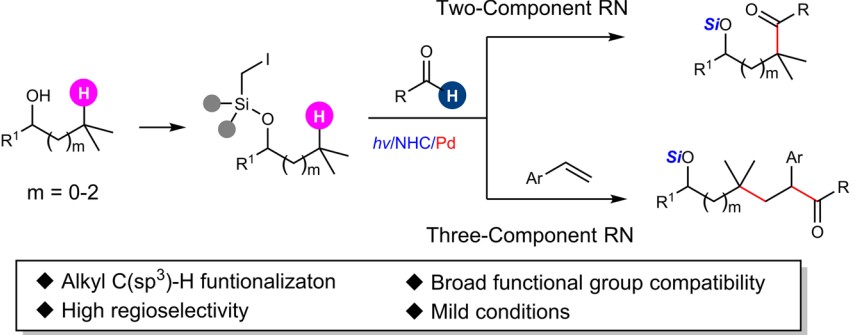

**d This Work: via Photoredox NHC/Pd-catalyzed Acylation with Alkyl C(sp³)-H**

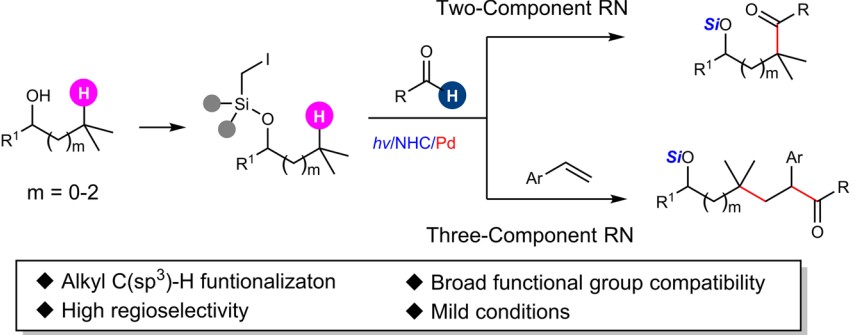

◆ Alkyl C(sp³)-H funtionalizaton   ◆ Broad functional group compatibility
◆ High regioselectivity            ◆ Mild conditions

**Fig. 1 | Ketone synthesis. a** Classical ketone synthesis from alcohols and acyl electrophiles. **b** Ketone synthesis via TM-catalyzed coupling with aldehydes. **c** Ketone synthesis via NHC-catalyzed acylation of activated alkylation reagents. **d** This work: ketone synthesis via photoredox NHC/Pd-catalyzed acylation of alkyl C(sp³)-H. X = halide. TM = transition metal. NHC = N-heterocyclic carbene. NHPI = N-hydroxyphthalimide. $hv$ = photoirradiation. RN = reaction.

the C–C bonds formation[34–37], including the synthesis of ketones from aldehydes (Fig. 1c). In 2019, Ohmiya and co-workers pioneered the NHC-catalyzed synthesis of ketones via the decarboxylative coupling of N-hydroxyphthalimide (NHPI) esters with aldehydes[38]. Lately, other single electron acceptors, such as Katritzky salts[39], oxime esters[40] or ethers[41], Togni reagents[42,43], and disulfides[44], were successfully employed for the synthesis of ketones with a similar strategy. In addition, the synthesis of ketone from acyl electrophiles under cooperative photoredox/NHC catalysis has also been reported[45–50].

The efficient and selective functionalization of C(sp³)-H bond has been a long-standing challenge in organic chemistry due to the intrinsic chemical inertness and similarity in electronic and steric properties of C(sp³)-H bonds[51,52]. The NHC-catalyzed acylation of C(sp³)-H bonds has broad potential application in ketone synthesis with reductive in the number of the synthetic steps, but is not well developed. The primary progress has been achieved in the acylation of activated benzylic and α-amino C(sp³)-H bonds by Rovis[53], Studer[54], Wang[55], and Ohmiya groups[56]. In contrast, the NHC-catalyzed acylation of ubiquitous unactivated alkyl C(sp³)-H bonds remains a considerable challenge[57]. Very recently, Li and co-workers reported NHC-catalyzed remote C(sp³)-H acylation of benzylic and unactivated alkyl C(sp³)-H bonds (only two special examples) of amides through N-centered radical-mediated 1,5-hydrogen atom transfer (HAT) mechanism[58].

In this work, we report the photoredox cooperative NHC/Pd-catalyzed synthesis of ketones via two and three-component reactions with alkyl C(sp³)-H functionalization (Fig. 1d). This process involves palladium-triggered primary carbon-centered radical-mediated 1,n-HAT to activate remote C(sp³)-H bonds[59–61] and the following coupling of the resultant radicals[62]. This reaction features mild reaction conditions and broad functional group tolerance. The potential application of this methodology is further illustrated by the late-stage acylation of the natural products.

## Results

### Condition optimization of two-component reaction

The two-component model reaction of iodomethylsilyl ether of aliphatic alcohol **1a** and picolinaldehyde **2a** was investigated under photoredox NHC/Pd catalysis (Table 1). We were encouraged that the

**Table 1 | Condition optimization[a]**

| Entry | preNHC | [Pd] | Ligand | 3a[b] (%) | 4a[c] (%) |
|---|---|---|---|---|---|
| 1 | N1 | Pd(OAc)$_2$ | L1 | 24 | 29 |
| 2 | N2 | Pd(OAc)$_2$ | L1 | 37 | 0 |
| 3 | N3 | Pd(OAc)$_2$ | L1 | 13 | 15 |
| 4 | N2 | Pd(OAc)$_2$ | L2 | 26 | 0 |
| 5 | N2 | Pd(OAc)$_2$ | L3 | 67(69[d]) | 0 |
| 6 | N2 | Pd(PPh$_3$)$_4$ | L3 | 39 | 0 |
| 7 | N2 | Pd(TFA)$_2$ | L3 | 19 | 14 |
| 8 | N2 | PdCl$_2$ | L3 | 17 | 19 |
| 9 | N2[e] | Pd(OAc)$_2$ | L2 | 13 | 0 |
| 10 | N2 | Pd(OAc)$_2$[f] | L3 | 29 | 19 |
| 11 | N2 | Pd(OAc)$_2$ | L3[g] | 20 | 11 |
| 12[h] | N2 | Pd(OAc)$_2$ | L3 | 29 | 0 |
| 13 | / | Pd(OAc)$_2$ | L3 | 0 | / |
| 14 | N2 | / | L3 | 0 | / |
| 15[i] | N2 | Pd(OAc)$_2$ | L3 | 0 | / |

[a]Reaction conditions: **1a** (0.1 mmol), **2a** (1.5 equiv.), [Pd] (10 mol%), Ligand (20 mol%), preNHC (20 mol%), Cs$_2$CO$_3$ (2.0 equiv.), 1.0 mL PhCF$_3$, 36 W Blue LEDs, r.t., under N$_2$.
[b]Yields of **3a** determined by $^1$H NMR using CH$_2$Br$_2$ as standard.
[c]Yields of **4a** determined by $^1$H NMR using CH$_2$Br$_2$ as standard.
[d]Isolated yields.
[e]10 mol% of preNHC was used.
[f]5 mol% of Pd(OAc)$_2$ was used.
[g]10 mol% of **L3** was used.
[h]36 w White LEDs.
[i]In dark. r.t. = room temperature.

reaction gave the desired acylation product **3a** after 1,6-HAT in 24% yield with similar amount of undesired direct coupling product **4a** and other byproducts[59] via dehydrogenation, when carried out in the presence of 20 mol% of thioazolium preNHC **N1**, 10 mol% of Pd(OAc)$_2$, 20 mol% of bidentate diphenyl-Xantphos **L1** with 2.0 equivalent Cs$_2$CO$_3$ in trifluorotoluene under blue LED irradiation (entry 1). Screening of preNHCs found that the preNHC **N2** with free hydroxyl group performed better to give ketone **3a** in 37% yield without **4a** (entry 2). When the preNHC **N3** with hydroxyl protected was used instead of **N2**, the yield of **3a** decreased and with the formation of **4a** (entry 3). Screening of ligands revealed that the reaction using dicyclohexyl-Xantphos **L2** instead of diphenyl **L1** gave the desired product **3a** in some decreased yield but without **4a** (entry 4). Thus, the monodentate tricyclohexylphosphine was then employed, resulting in dramatical improvement of the yield of **3a** without **4a** (entry 5). Other palladium source, such as Pd(PPh$_3$)$_4$, Pd(TFA)$_2$, and PdCl$_2$ is inferior to Pd(OAc)$_2$ (entries 6–8). Decreasing the loading of preNHC, Pd, or ligand resulted in decreased yields (entries 9–11). The yield of **3a** was decreased to 29% under white LEDs irradiation (entry 12). Control experiments revealed all of preNHC, Pd and photoirradiation are crucial for the reaction (entries 13–15).

## Substrate scope of two-component reaction

With the optimized reaction conditions in hand, the substrate scope of aldehyde was then explored (Figs. 2 and 3a–ah). Picolinaldehyde with 3-chloro resulted in decreased yield (**3b**), possible due to difficulty of the formation of Breslow intermediate stemming from the hindrance of the chloro substituent. The reaction of picolinaldehyde with different substituents, such as Me, OMe and Cl, at 4 or 5-position, all reacted smoothly to give ketones (**3c–3g**) in moderate to good yields. 6-Methyl substituent on picolinaldehyde resulted in low yield (**3h**), possibly owing to the increasing steric hindrance. Alkynyl substituent was well tolerated, giving ketone **3i** in 90% yield. The picolinaldehydes with a range of additional substituted aryl group (Ar' = 4-MeOC$_6$H$_4$, 4-CNC$_6$H$_4$, 3,5-(CF$_3$)$_2$C$_6$H$_3$), heteroaryl group (Ar' = 2-thienyl) and fused aryl group (Ar' = 1-naphthyl, 9-anthranyl, 9-phenanthryl), all worked well, providing the corresponding products **3j-3q** in moderate to good yields. Isoquinoline-3-carbaldehyde performed much better than quinoline-2-carbaldehyde (**3r** vs. **3s**), possibly due to the increasing steric hindrance. Pyrazine-2-carbaldehyde afforded the desired product (**3t**) in 47% yield. The reaction of thiazole-2-carbaldehydes furnished the corresponding products **3u-3v** in decreased yields.

Unexpectedly, benzaldehyde did not work under the same conditions as picolinaldehyde. Considering the possible coordination of palladium with picolinaldehyde, 2-hydroxypyridine derivatives were added as additive for the reaction of benzaldehyde (see Supplementary Table 3). We were pleased to find that the reaction of benzaldehyde with aliphatic silyl ether **1a** afforded multifunctionalized tetrahydrofuran **3w'** in 61% yield after desilylation, when 20 mol% of 2-hydroxy-5-trifluoromethylpyridine was added as the additive. The corresponding reaction with γ-phenyl silyl ether proceeded smoothly to produce **3x** in 72% yield with acylation of benzylic C(sp$^3$)-H bond. The scope with respect to benzaldehyde was evaluated. Benzaldehydes bearing electron-donating (X = 4-Me) or electron-withdrawing groups (X = 4-F, 4-Cl, 4-Br, 4-CF$_3$, 4-CO$_2$Me, 4-CN) on the *para*-position all worked well to give the corresponding ketones **3y-3ae** in moderate to good yields. The benzaldehydes with *meta*- or *ortho*-substituent (3-Cl, 2-CN) were also tolerated affording the products **3af** and **3ag** in 48% and 88% yield, respectively. In addition, cinnamyl aldehyde was also compatible to produce the α,β-unsaturated ketone **3ah** albeit with decreased yield. However, simple aliphatic aldehydes did not work for the reaction under the current reaction conditions.

Silyl ether derived from varied alcohol was then examined (Figs. 2 and 3ai–az). Primary alcohols, such as 3-methylbutan-1-ol and

2-cyclohexylethan-1-ol, furnished the desired ketones **3ai** and **3aj** in good yields. It was noteworthy that the reaction of non-symmetric secondary alcohol bearing two tertiary C(sp$^3$)-H gave the ketone **3ak** with exclusive γ-regioselectivity via 1,6-HAT. The terminal chloro in secondary alcohol was tolerated, affording the ketone **3al** in 67% yield. Interestingly, hexan-2-ol and cyclooctanol derived ether without tertiary C(sp$^3$)-H could also participate in the reaction with exclusive γ-regioselectivity at secondary C(sp$^3$)-H albeit in low yields (**3am** & **3an**). Beyond the favored 1,6-HAT, the reaction of alkyl ether with β- and δ-tertiary C(sp$^3$)-H gave the corresponding ketones **3ao** and **3ap** in acceptable yields via 1,5- and 1,7-HAT, respectively. 4-Arylbutan-2-ols bearing different substituents (X = 4-H, 4-OMe, 4-NMe$_2$, 4-Br, 2-Br) all worked well to give the corresponding ketones **3aq-3au** in moderate to good yields. The reaction of iodomethylsilyl ether of 1-phenylpropan-2-ol afforded the corresponding product **3av** via 1,5-HAT in 94% yield. The silyl ether with γ-benzyloxyl worked for the reaction, giving the corresponding product **3aw** with C-H acylation at the alpha-position of the benzyloxy in 76% yield. In addition, ester was tolerated to deliver the corresponding product **3ax** in 63% yield. Importantly, cholesterol and dehydroepiandrosterone worked well for the reaction, yielding the corresponding ketones **3ay** and **3az** in good yields, which showed potential application of this methodology in late-stage functionalization.

## Substrate scope of three-component reaction

The carboacylation of alkenes provide easy access to ketones. Recently, a variety of groups have developed TM-catalyzed intramolecular[63,64] or intermolecular carboacylation[65–68] with acyl electrophiles. The NHC-catalyzed carboacylation of alkenes using aldehydes was also reported[57,62]. However, these reactions require active alkylation reagents as radical precursor. Following the two-component photoredox NHC/Pd-catalyzed C(sp$^3$)-H acylation reaction, we then tested the corresponding three-component reaction with the addition of styrenes (Fig. 3). We were happy to find that the three-component reaction went smoothly under same conditions as the two-component reaction, and the corresponding ε-hydroxylketones **6** were obtained in good to high yields via alkylacylation of styrenes and following desilylation with tetra-butylammonium fluoride (TBAF).

The reaction of picolinaldehydes with different substituents at varied position (4-Cl, 4-OMe, 5-Me, 5-OMe, 5-Cl and 6-OMe) of the pyridine ring, went smoothly to give ε-hydroxylketones (**6a–6g**) in good to high yields. 4-Alkynylpicolinaldehyde was also tolerated, giving the ketone (**6h**) with multifunctional groups in 79% yield. Picolinaldehydes with varied additional 5-aryl groups (Ar'= C$_6$H$_5$, 4-OMeC$_6$H$_4$, 3,5-(CF$_3$)$_2$C$_6$H$_3$) or fused aryl groups (Ar' = 1-naphthyl, 9-anthranyl, 9-phenanthryl) all worked well to afford the corresponding ε-hydroxylketones (**6i-6n**) in good to high yields. Several other heteroaryl aldehydes, such as quinoline-2-carbaldehyde, isoquinoline-3-carbaldehyde and thiazole-4-carbaldehyde, performed as well, providing the ε-hydroxylketones (**6o-6q**) in good yields.

A variety of styrenes were then tested for the reaction. It is worth noting that all styrenes with both electron-donating (X = OMe, $^t$Bu) and electron-withdrawing (X = F, Cl, Br, CN, CF$_3$) groups at the *para*-position worked well, leading to ε-hydroxyl **6r-6x** in good yields. Substituents at the *meta*-, *ortho*-position were tolerated to afford the products **6y-6z** in high yields. The reaction of 2-vinylnaphthalene gave ketone **6aa** in 69% yield. The reaction with 2-vinylpyridine or 4-vinylpyridine produced ketones **6ab** and **6ac** in good yields. Notably, the reaction of 1,1-diphenyl-buta-1,3-diene with conjugated C=C bonds showed exclusive regioselectivity (**6ad**), giving only the 1,2-alkylacylation of the terminal C=C bond in 98% yield. ω-Chlorine of the alkyl silyl ether was tolerated for the reaction, giving the ketone **6ae** in 58% yield. In addition, the

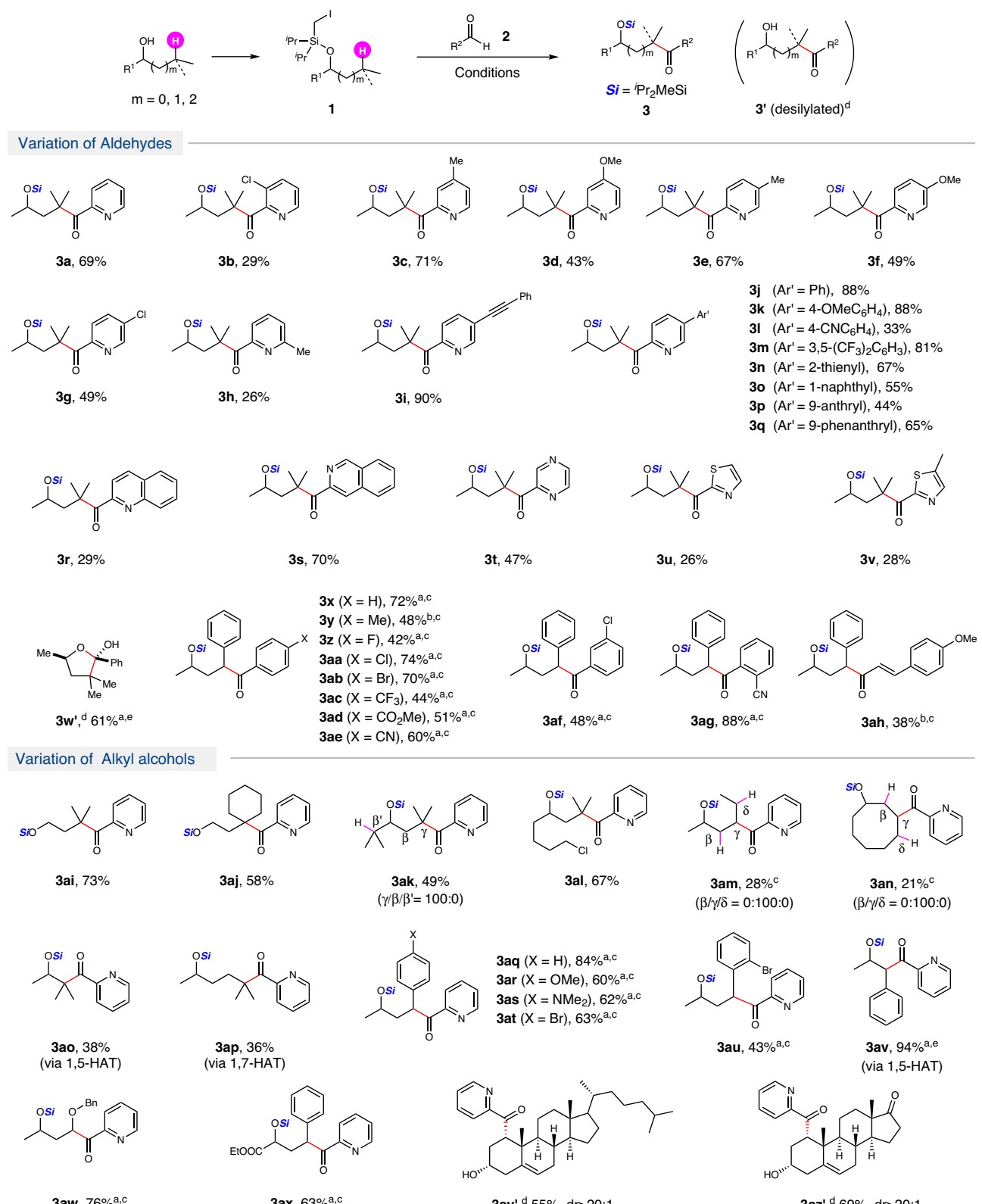

**Fig. 2 | Two-component reaction.** Conditions: **1** (0.3 mmol), **2** (1.5 equiv.), Pd(OAc)$_2$ (10 mol%), PCy$_3$ (20 mol%), **N2** (20 mol%), Cs$_2$CO$_3$ (2.0 equiv.), 3.0 mL PhCF$_3$, 36 W Blue LEDs, r.t., under N$_2$, 16 h; [a]with 5-(trifluoromethyl)pyridin-2-ol (20 mol%); [b]with 4-methylpyridin-2-ol (20 mol%); [c]dr = 1:1; [d]After treating with TBAF (2.0 equiv.), 2 h.

reaction of menthol derived silyl ether gave the corresponding product **6af** with exclusive regioselectivity at the tertiary carbon of the isopropyl group even in the presence of two other tertiary C(sp³)-H. However, the reaction with internal or aliphatic olefins gave no desired ketone products under current conditions.

To verify the practicality of this protocol, gram-scale experiments and further chemical transformations of the products were carried out (Fig. 4). The reaction of alkyl silyl ether **7a** from alcohol **7** (1.07 g, 6 mmol) with picolinaldehyde via γ-C(sp³)-H acylation under the standard conditions gave the silyl ether of chloroketone **3z**,

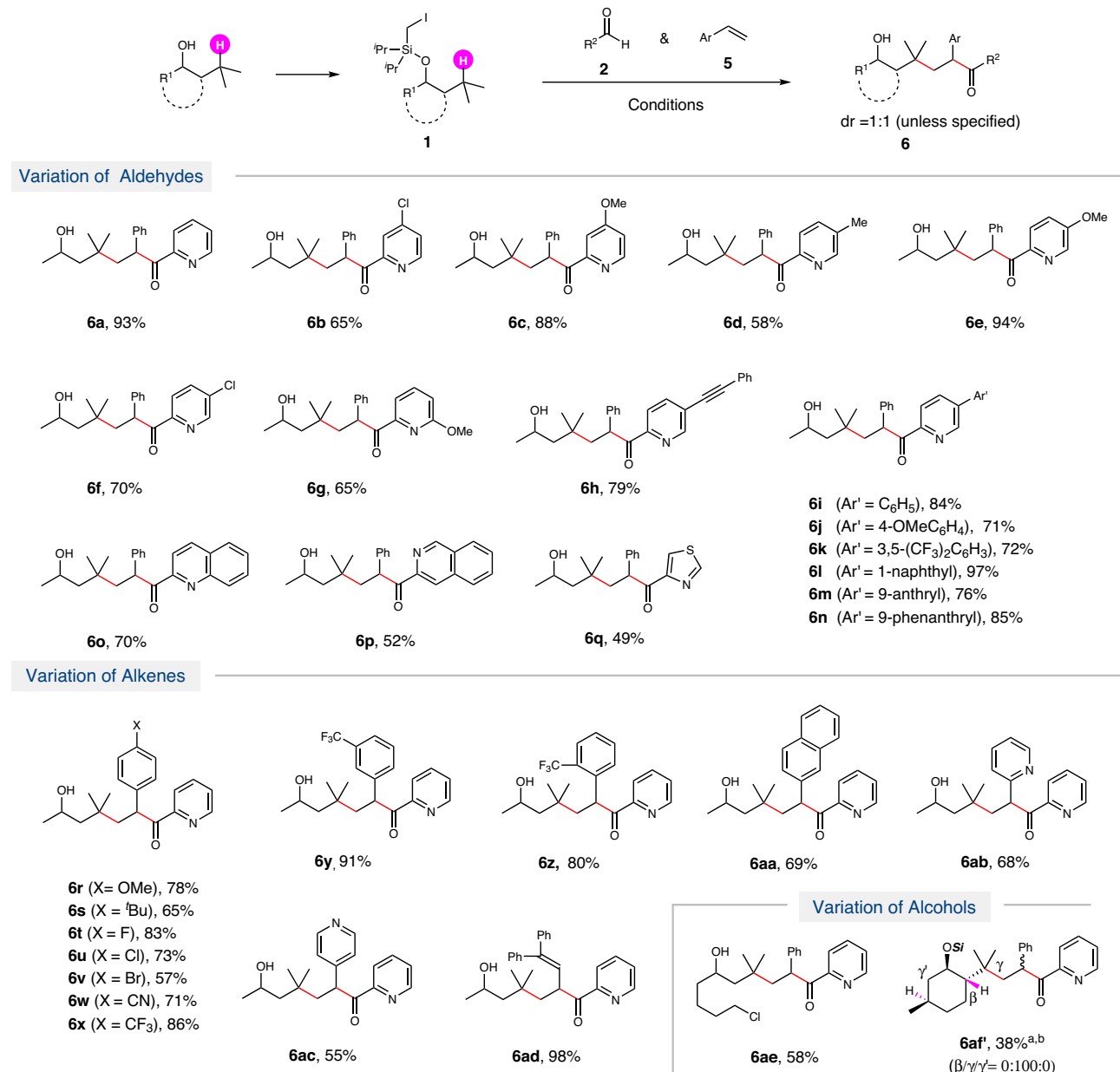

**Fig. 3 | Three-component reaction.** Conditions: **1** (0.3 mmol), **5** (2.0 equiv.), **2** (1.5 equiv.), Pd(OAc)$_2$ (10 mol%), PCy$_3$ (20 mol%), **N2** (20 mol%), Cs$_2$CO$_3$ (2.0 equiv.), 3.0 mL PhCF$_3$, 36 W Blue LEDs, r.t., under N$_2$, 16 h; then TBAF (2.0 equiv.); [a]Without TBAF; [b]dr = 5:1.

which was desilylated to give hemiketal **8** (1.01 g, 59% overall yield from **7**) of the corresponding γ-hydroxyl ketone. Reduction of hemiketal **8** with LiAlH$_4$ gave the corresponding 1,4-diol **9** in 56% yield. The reaction of ω-chlorohemiketal **8** with TMSCN afforded the corresponding silylated cyanohemiketal **10** in 55% yield. In addition, 1.12 g of ε-hydroxyl ketone **6a** was also obtained from 4-methylpentan-2-ol via the process of silylation, three-component coupling with styrene and picolinaldehyde, and desilylation (60% overall yield from alcohol). Dess-Martin oxidation of ε-hydroxyl ketone **6a** delivered diketone **11** in 79% yield.

## Discussion
### Mechanistic studies
A series of control experiments were carried out, in order to investigate the mechanism of this reaction. No acylation product **3a** was detected when PhSeSePh was added as a radical scavenger, while the adduct **12** from the alkyl radical and PhSeSePh was detected by

HRMS and isolated in 43% yield (Fig. 5a). These results indicated that the existence of primary alkyl radical via dehalogenation of the iodomethylsilyl ether. Crossover experiment with (iodomethyl)tri-methylsilane as radical precursor and silyl ether **1a'** without the iodomethyl substituent as proton source of possible intermolecular HAT was carried out, which resulted in no formation of ketone **14** via intermolecular HAT but ketone **13** in 73% yield without the partici-pation of silyl ether **1a'** (fully recovered) (Fig. 5b). This result rules out the intermolecular HAT pathway and suggests the formation of tertiary/secondary alkyl radical from primary alkyl radical via the intramolecular 1,n-HAT for the titled reaction. In addition, the reaction could not be fully inhibited by the addition of D$_2$O or MeOH to trap the possible carbon cation or anion (Fig. 5c), which support the radical coupling pathway instead of radical/polar crossover pathway for the reaction.

The light on/off experiments revealed that blue light was essential for this reaction and the chain reaction process was

**Fig. 4 | Gram-scale synthesis and further chemical transformations. a** Si reagent (1.5 equiv.), imidazole (2.0 equiv.), THF, r.t. **b** Picolinaldehyde (1.5 equiv.), Pd(OAc)₂ (10 mol%), PCy₃ (20 mol%), **N2** (20 mol%), Cs₂CO₃ (2.0 equiv.), PhCF₃, 36 W Blue LEDs, r.t. **c** TBAF (2.0 equiv.), THF, r.t. **d** LiAlH₄ (2.0 equiv.), Et₂O, 0 °C. **e** TMSCN (3.0 equiv.), KF (3.0 equiv.), DMF, 80 °C. **f** (1) Si reagent (1.5 equiv.), imidazole (2.0 equiv.), THF, r.t.; (2) Styrene (2.0 equiv.), picolinaldehyde (1.5 equiv.), Pd(OAc)₂ (10 mol%), PCy₃ (20 mol%), **N2** (20 mol%), Cs₂CO₃ (2.0 equiv.), PhCF₃, 36 W Blue LEDs, r.t.; (3) TBAF (2.0 equiv.), THF, r.t. **g** Dess-Martin periodinane (2.0 equiv.), CH₂Cl₂, r.t.

excluded (see Supplementary Fig. 1). The UV−visible absorption spectra of the starting materials and catalysts were measured (see Supplementary Fig. 2). It was found that no apparent absorption for the alkyl silyl ether **1a**, styrene **5a**, picolinaldehyde **2a**, carbene **N2**, phosphine ligand PCy₃ observed. There is weak absorption for Pd(OAc)₂, which could be dramatically enhanced by the addition of phosphine ligand and/or NHC. These results suggest that Pd species worked as photocatalysts.

Based on the mechanistic experiments and previous works[59,62], the plausible catalytic cycle of this reaction was shown in Fig. 6. The Pd(0)L complex undergoes excitation by blue light to form the active [Pd(0)L]* catalyst. The single electron transfer (SET) between the [Pd(0)L]* catalyst and iodomethylsilyl ether **1** generates Pd(I) complex and methyl radical **I** with the leaving of iodide. The following intramolecular 1,n-HAT (*n* = 5, 6, 7) of methyl radical **I** produces secondary/tertiary radical species **II**. Meanwhile, the Breslow intermediate anion **III**, generated from aldehyde **2** under **NHC** catalysis, is single electron oxidized by Pd(I) complex to afford persistent ketyl radical **IV** and regenerates Pd(0)L catalyst. The coupling of persistent ketyl radical **IV** and transient alkyl radical **II** gives the adduct **V** of the two-component reaction, which is fragmented to afford the C(sp³)-H acylation product **3** and release **NHC** catalyst (two-component reaction).

When styrene **5** is added, the addition of alkyl radical **II** to styrene **5** affords more stable benzylic radical **VI**, which is then coupled with ketyl radical **IV** to furnish the three-component adduct **VII**. The fragmentation of adduct **VII** gives the alkylacylation product **6'** and regenerates **NHC** catalyst (three-component reaction).

In summary, this work demonstrates the generation of ketyl radical and alkyl radical under the photoredox cooperative NHC/Pd catalysis, and provides two and three-component reactions for the synthesis of ketones via unactivated alkyl C(sp³)-H functionalization with aldehydes as the acylation reagents. The two-component reaction of iodomethylsilyl alkyl ether with aldehydes gave a variety β-, γ- and δ-silyloxylketones via 1,n-HAT (*n* = 5, 6, 7) of silylmethyl radicals to generate secondary or tertiary alkyl radicals and following coupling with ketyl radical from aldehydes under photoredox NHC catalysis. The three-component reaction with the addition of styrenes gave the corresponding ε-hydroxylketones via the generation of benzylic radical by the addition of alkyl radical to styrenes and following coupling with ketyl radical. Further investigation on photoredox cooperative NHC/TM catalysis is underway in our laboratory.

## Methods

### General procedure for two-component reaction

A 5 mL vial equipped with a stir bar was charged with **N2** (0.06 mmol), Pd(OAc)₂ (0.03 mmol), PCy₃ (0.06 mmol), additive (0.06 mmol for benzaldehydes and enals) and 2.0 mL of PhCF₃. After stirring for 30 min in glove box, to the solution was added Cs₂CO₃ (0.6 mmol), aldehydes **2** (0.45 mmol), alkyl silyl ethers **1** (0.3 mmol), and 1.0 mL of PhCF₃. The reaction mixture was removed from the glove box and stirred under 36 W blue LED lights at room temperature for 16 h. The solution was concentrated under reduced pressure, and purified by column chromatography on silica gel to afford the desired ketones **3**.

### General procedure for three-component reaction

A 5 mL vial equipped with a stir bar was charged with **N2** (0.06 mmol), Pd(OAc)₂ (0.03 mmol), PCy₃ (0.06 mmol) and 2.0 mL of PhCF₃. After stirring for 30 min in glove box, to the solution was added Cs₂CO₃ (0.6 mmol), aldehydes **2** (0.45 mmol), styrenes **5** (0.6 mmol), alkyl silyl ethers **1** (0.3 mmol), and 1.0 mL of PhCF₃. The reaction mixture was removed from the glove box and stirred under 36 W blue LED lights at room temperature for 16 h. Then, TBAF (0.6 mmol, 1.0 M in THF) was added. After being stirred for 2 h, the solution was concentrated under reduced pressure, and purified by column chromatography on silica gel to afford the desired ketones **6**.

**a Radical Trapping Experiment**

1a + 2a →[Standard Conditions / PhSeSePh (10 equiv)] **12**, 43% + **3a**, N.D.

**b Crossover Experiment**

1a' + 5a + 2a →[Standard Conditions / TMS–I (1.5 equiv)] **13**, 73% + **14**, N.D.

**c Cation and Anion Trapping Experiment**

1a + 2a →[Standard Conditions / D₂O (10 equiv)] **3a**, 19% + N.D.

1a + 2a →[Standard Conditions / MeOH (10 equiv)] **3a**, 42% + N.D.

**Fig. 5 | Mechanistic experiments. a** Radical trapping experiment with diphenyl diselenide. **b** Crossover experiment with the addition of (iodomethyl)trimethylsilane. **c** Cation and anion trapping experiment with the addition of deuterium oxide or methanol. N.D. not detected.

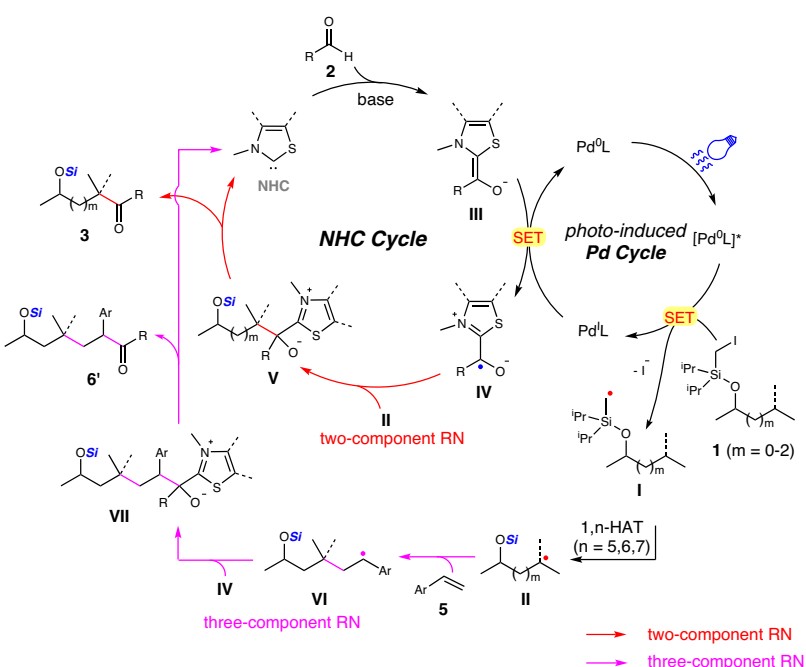

**Fig. 6 | Proposed mechanism.** The plausible mechanism for cooperative NHC/Pd-catalyzed functionalization of remote C(sp³)-H bond of alcohols.

## Data availability

The authors declare that the data supporting the findings of this study are available within the article and its Supplementary Information file. For experimental details and compound characterization data see Supplementary Methods. For $^1$H NMR, $^{13}$C NMR spectra see Supplementary Figs. 3–191.

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

## Acknowledgements

Financial support from the National Natural Science Foundation of China (Nos 21831008, S.Y.; 22271292, C.-L.Z.) and Beijing National Laboratory for Molecular Sciences (BNLMS-CXXM-202003, S.Y.) and the Ministry of Science and Technology of China is greatly acknowledged.

## Author contributions

H.-Y.W., X.-H.W., B.-A.Z., C.-L.Z., and S.Y. designed, performed, and analyzed the experiments. H.-Y.W., C.-L.Z., and S.Y. co-wrote the manuscript. H.-Y.W., C.-L.Z., and S.Y. contributed to the discussions.

## Competing interests

The authors declare no competing interests.
