## [Peer Review File · Nature Communications]

REVIEWER COMMENTS

Reviewer #1 (Remarks to the Author):

Zhang and Ye present a novel method for the synthesis of ketones from aldehydes using a combination of catalytic systems. This approach is particularly interesting because it involves the photocatalytic Pd-catalyzed activation of alkyl C(sp³)-H bonds in corporation with NHC catalysis. However, the scope of the aldehydes used in this transformation is limited to picolinaldehydes. These restrictions could potentially be an issue and have not been seen in previous NHC-catalyzed ketone syntheses.

Due to the cutting-edge quality of Nature Communications, this work may be published in the current form. However, if the scope of aldehydes could be expanded to include common aromatic aldehydes and alkyl aldehydes, or if the synthesis of enantiorich products could be achieved with the use of chiral NHC catalysts, the work could be considered for publication in Nature Communications as an article.

1) The synthesis of ketones through the NHC-catalyzed radical pathway has already been reported by many research groups. Although the 1,6-H migration strategy based on Pd-catalysis has not been reported, the authors need to elaborate systematically on the significance of this work in the field and highlight the potential applications of this methodology in the synthesis of complex molecules. In the meantime, the approaches towards known compounds with similar structures should be presented to demonstrate their advantages in synthesis.

2) The authors need to incorporate more diverse substrates in their research to showcase the versatility of the method. Currently, the alkyl parts of the substrates are overly simplified, with limited functional group substitutions. The authors should present a wider range of examples with functional groups such as NH₂-, OH-, ketone, CN-, ester, etc. A variety of backbone structures are needed, such as benzylic position, heteroatoms at the alpha position, primary and secondary carbons. This would make the schemes much more compelling.

3) Reports on related NHC-ketone formation should be cited:

Org. Lett. 2022, 24, 5519–5524; ACS Catal. 2022, 12, 24, 15241–15248; J. Am. Chem. Soc. 2022, 144, 22767–22777.

4) The authors should testify the use of chiral NHC ligands to obtain enantiorich products under similar conditions.

Reviewer #2 (Remarks to the Author):

The current manuscript by Ye and co-workers described a photoredox cooperative NHC/Pd catalyzed alkyl C(sp³)-H acylation that allowed the synthesis of ketones from aldehydes. The access of ketones from aldehydes is a high atom- and step-economic method. Previously, the NHC-catalyzed C(sp³)-H acylation relied on the acylation of benzylic and α -amino C(sp³)-H bonds, while the acylation of unactivated alkyl C(sp³)-H is challenging and there are only two special examples can be documented (Angew. Chem. Int. Ed. 2022, 61, e202116629). In this work, the author utilizes a photoredox cooperative NHC/Pd-catalysis for the access of a variety of β -, γ - and δ -hydroxyketones from aldehydes and iodomethylsilyl alkyl ether, which proceed via 1,n-HAT (n = 5, 6, 7) of silylmethyl radicals to generate secondary or tertiary alkyl radicals and following coupling with ketyl radicals from aldehydes. A series of mechanistic experiments were also conducted that well supported the proposed reaction pathway. Besides, a three-component reaction with styrenes was further developed that could allow the synthesis of ϵ -hydroxyketones. In addition, the power of the method was also highlighted by the late-stage functionalization of natural products. Overall, this is a very nice work and publication for Nature Communications after the issues itemized below have been addressed is recommended.

- 1) Did the authors screen other types of NHC precursors, such as imidazolium or thiazolium? This is important for readers to understand how the authors realized this reaction.
- 2) The scope with respect to aldehydes seems to be restricted to 2-heteroaromatic ones. How is the reaction going when aliphatic aldehydes or benzaldehydes are employed? The limitation of this method should be discussed.
- 3) In the three-component reaction, how is the reaction going if internal or aliphatic are employed?
- 4) The d.r. value of compounds 3ae and 3af should be provided.
- 5) This paper should be cited (J. Am. Chem. Soc. 2021, 143, 4903)

Reviewer #3 (Remarks to the Author):

In this manuscript, Ye and co-workers develop an interesting cooperative NHC/Palladium-catalyzed synthesis of ketones from aldehydes via alkyl C(sp³)-H acylation. Catalytic direct synthesis of ketones from aldehydes via C-H functionalization is an attractive approach with high step-economy. The strategy involves a silylmethyl radical-mediated 1,n-HAT to generate secondary/tertiary alkyl radicals followed by trapping with ketal radicals (for two-component reaction), affording γ -hydroxyl ketone derivatives in moderate to good yields with predictable regioselectivity. By the addition of styrenes, three-component reaction was realized via radical relay process, leading to ϵ -hydroxyl ketones in moderate to high yields. The scope of substrates has been well evaluated with various aldehydes, iodomethylsilyl ether, and styrenes. Gram-scale reactions and late-stage acylation of natural and bioactive molecules prove the potential practicability of this reaction. The manuscript is

well presented. The publication in Nature Communications is recommended after addressing some minor issues.

1. What if the reaction is conducted under other wavelength LEDs irradiation?

2. How about using other aldehydes, such as aliphatic ones?

3. Beyond styrenes, what about another type of olefins?

4. Some typos, such as “idomethylsilyl”, “varirety”, “intarmolecular”...

5. The characterization data are suggested to be carefully checked. In many cases, identical values are presented in the ^{13}C NMR data. Please replace them as two-digit after the decimal point.

Responses to Reviewer Comments

Reviewer 1:

[Q1]: However, the scope of the aldehydes used in this transformation is limited to picolinaldehydes. These restrictions could potentially be an issue and have not been seen in previous NHC-catalyzed ketone syntheses.

[R1]: Thanks. We have successfully expanded scope of the reaction to benzaldehydes beyond picolinaldehydes. Considering the possible coordination of palladium with picolinaldehyde, 2-hydroxypyridine derivatives were added as additive for the reaction of benzaldehydes (see Supplementary Table S3). The variation of benzaldehydes has been added in Fig.2 (3w'-3ag). In addition, cinnamyl aldehyde was also compatible (3ah) albeit with decreased yield.

[Q2]: Although the 1,6-H migration strategy based on Pd-catalysis has not been reported, the authors need to elaborate systematically on the significance of this work in the field and highlight the potential applications of this methodology in the synthesis of complex molecules. In the meantime, the approaches towards known compounds with similar structures should be presented to demonstrate their advantages in synthesis.

[R2]: Thanks. The introduction has been revised. The primary progress has been achieved in the acylation of activated benzylic and α -amino $C(sp^3)$ -H bonds by Rovis, Studer, Wang, and Ohmiya groups. Very recently, Li and co-workers reported NHC-catalyzed remote $C(sp^3)$ -H acylation of benzylic and unactivated alkyl $C(sp^3)$ -H bonds (only two special examples) of amides through N-centered radical-mediated 1,5-hydrogen atom transfer (HAT) mechanism. In this work, we report the photoredox cooperative NHC/Pd-catalyzed acylation of unactivated alkyl $C(sp^3)$ -H bonds via palladium-triggered primary carbon-centered radical-mediated 1,n-HAT to activate remote $C(sp^3)$ -H bonds. The potential application of this methodology

was further illustrated by the late-stage acylation of the natural products.

[Q3] The authors should present a wider range of examples with functional groups such as NH₂-, OH-, ketone, CN-, ester, etc.

[R3]: Thanks. The substrates with functional groups such as OMe, NMe₂, Br, (Fig. 2 Compound 3aq-3au), ether, ester (3aw, 3ax) have been successfully employed for the reaction.

[Q4] A variety of backbone structures are needed, such as benzylic position, heteroatoms at the alpha position, primary and secondary carbons.

[R4]: The substrates with reaction at the benzylic position (Fig. 2 compound 3aq-3av), at alpha position of heteroatom (3aw), secondary carbons (3am-3an) have been successfully employed. However, acylation at primary C-H failed under current condition possibly due to instability of primary radicals.

[Q5]: Reports on related NHC-ketone formation should be cited:

Org. Lett. 2022, 24, 5519–5524; ACS Catal. 2022, 12, 24, 15241–15248; J. Am. Chem. Soc. 2022, 144, 22767–22777.

[R5]: Thanks. These reports have been cited as ref. 44, 49, and 41.

[Q6]: The authors should testify the use of chiral NHC ligands to obtain enantiorich products under similar conditions.

[R6]: We tried to develop the asymmetric variant of this reaction with the use of chiral preNHCs, but failed to obtain the desired products under current conditions (see Supplementary Table S4).

Reviewer 2:

[Q7]: Did the authors screen other types of NHC precursors, such as imidazolium or thiazolium? This is important for readers to understand how the authors realized this reaction.

[R7]: Thanks. During the optimization of the reaction conditions, other types of NHC precursors such as imidazolium and triazolium, were also evaluated, but unefficient in this reaction. The reaction with thiazolium-based NHC precursors could afford the desired ketone products in varied yields. The optimization of preNHCs has been added to revised Supplementary as Table S1.

[Q8]: The scope with respect to aldehydes seems to be restricted to 2-heteroaromatic ones. How is the reaction going when aliphatic aldehydes or benzaldehydes are employed? The limitation of this method should be discussed.

[R8]: Thanks. We have successfully expanded scope of the reaction to benzaldehydes. The variation of benzaldehydes has been added in Fig. 2 (3w'-3ag). In addition, cinnamyl aldehyde was also compatible (3ah) albeit with decreased yield. However, simple aliphatic aldehydes did not work for the reaction under the current reaction conditions.

[Q9]: In the three-component reaction, how is the reaction going if internal or aliphatic are employed?

[R9]: Thanks. When internal or aliphatic olefins were employed, the reaction did not work under current conditions.

Unsuccessful Alkenes

[Q10]: The d.r. value of compounds 3ae and 3af should be provided.

[R10]: Thanks for your suggestion. Only sole diastereoisomer (d.r. > 20:1) was

obtained for compounds 3ay' and 3az' (3ae and 3af in previous manuscript). It has been added in the revised manuscript as note in Fig. 2 (3ay', 3az').

[Q11]: This paper should be cited (J. Am. Chem. Soc. 2021, 143, 4903)

[R11]: Thanks. This paper has been cited as ref. 50.

Reviewer 3:

[Q12]: What if the reaction is conducted under other wavelength LEDs irradiation?

[R12]: Thanks for your suggestion. The yield of **3a** was decreased to 29% under White LEDs irradiation (Table 1, entry 12).

[Q13]: How about using other aldehydes, such as aliphatic ones?

[R13]: We have successfully expanded scope of the reaction to benzaldehydes (Fig. 2 3w'-3ag) beyond previous picolinaldehydes. Cinnamyl aldehyde was also compatible (3ah) albeit with decreased yield. However, simple aliphatic aldehydes did not work for the reaction under the current reaction conditions.

[Q14]: Beyond styrenes, what about another type of olefins?

[R14]: Internal or aliphatic olefins did not work for the reaction under current conditions (For detail results, see response [R9])

[Q15]: Some typos, such as "idomethylsilyl", "variety", "intarmolecular"...

[R15]: Thanks. These typos have been revised accordingly.

[Q16]: 5. The characterization data are suggested to be carefully checked. In many cases, identical values are presented in the ¹³C NMR data. Please replace them as two-digit after the decimal point.

[R16]: Thanks. The characterization data have been checked carefully, and

the identical values have been revised accordingly.

Editor's

[Q17]: In particular, we suggest broadening the scope on the aldehydes as well as olefins, as suggested by our reviewers, and the reframing of the novelty in light of the current literature.

[R17]: Thanks. We have successfully broadened the scope on the aldehydes from picolinaldehydes to benzaldehydes (See also response, [R1]). Internal or aliphatic olefins did not work for the reaction under current conditions (See also response [R9]). The introduction has been revised accordingly (See also response [R2]).

REVIEWERS' COMMENTS

Reviewer #1 (Remarks to the Author):

All the issues have been addressed in the revision and it is recommended to accept the article in current form.

Reviewer #2 (Remarks to the Author):

This reviewer agrees to publish the article now.

One more suggestion, although the internal or aliphatic olefins did not work in the three-component reaction under the current conditions, the results are welcome to be added to the SI.

Reviewer #3 (Remarks to the Author):

All questions have been answered very well, so it is recommended to accept.

Responses to Reviewer Comments

Manuscript: NCOMMS-23-05471A

Reviewer #1 (Remarks to the Author):

Comment: All the issues have been addressed in the revision and it is recommended to accept the article in current form.

Response: Thanks.

Reviewer #2 (Remarks to the Author):

Comment: One more suggestion, although the internal or aliphatic olefins did not work in the three-component reaction under the current conditions, the results are welcome to be added to the SI.

Response: Thanks for your suggestion. The corresponding results have been added to the Supplementary Information, S41.

Reviewer #3 (Remarks to the Author):

Comment: All questions have been answered very well, so it is recommended to accept.

Response: Thanks.